# Classification of Brain Tumors from MRI Images Using a Convolutional Neural Network

**Milica M. Badža** [1,2,*] and **Marko Č. Barjaktarović** [2]

1    Innovation center, School of Electrical Engineering, University of Belgrade, Bulevar kralja Aleksandra 73, 11000 Belgrade, Serbia
2    School of Electrical Engineering, University of Belgrade, Bulevar kralja Aleksandra 73, 11000 Belgrade, Serbia; mbarjaktarovic@etf.bg.ac.rs
*    Correspondence: milica.badza@ic.etf.bg.ac.rs; Tel.: +381-11-321-8455

**Abstract:** The classification of brain tumors is performed by biopsy, which is not usually conducted before definitive brain surgery. The improvement of technology and machine learning can help radiologists in tumor diagnostics without invasive measures. A machine-learning algorithm that has achieved substantial results in image segmentation and classification is the convolutional neural network (CNN). We present a new CNN architecture for brain tumor classification of three tumor types. The developed network is simpler than already-existing pre-trained networks, and it was tested on T1-weighted contrast-enhanced magnetic resonance images. The performance of the network was evaluated using four approaches: combinations of two *10*-fold cross-validation methods and two databases. The generalization capability of the network was tested with one of the *10*-fold methods, subject-wise cross-validation, and the improvement was tested by using an augmented image database. The best result for the *10*-fold cross-validation method was obtained for the record-wise cross-validation for the augmented data set, and, in that case, the accuracy was 96.56%. With good generalization capability and good execution speed, the new developed CNN architecture could be used as an effective decision-support tool for radiologists in medical diagnostics.

**Keywords:** brain tumor classification; convolutional neural network; image classification; magnetic resonance imaging; machine learning; medical imaging; neural networks

## 1. Introduction

Cancer is the second leading cause of death globally, according to the World Health Organization (WHO) [1]. Early detection of cancer can prevent death, but this is not always possible. Unlike cancer, a tumor could be benign, pre-carcinoma, or malign. Benign tumors differ from malign in that benign generally do not spread to other organs and tissues and can be surgically removed [2].

Some of the primary brain tumors are gliomas, meningiomas, and pituitary tumors. Gliomas are a general term for tumors that arise from brain tissues other than nerve cells and blood vessels. On the other hand, meningiomas arise from the membranes that cover the brain and surround the central nervous system, whereas pituitary tumors are lumps that sit inside the skull [3–6]. The most important difference between these three types of tumors is that meningiomas are typically benign, and gliomas are most commonly malignant. Pituitary tumors, even if benign, can cause other medical damage, unlike meningiomas, which are slow-growing tumors [5,6]. Because of the information mentioned above, the precise differentiation between these three types of tumors represents a very important step of the clinical diagnostic process and later effective assessment of patients.

The most common method for differential diagnostics of tumor type is magnetic resonance imaging (MRI). However, it is susceptible to human subjectivity, and a large amount of data is

difficult for human observation. Early brain–tumor detection mostly depends on the experience of the radiologist [7]. The diagnostics of the tumor could not be complete before establishing whether it is benign or malignant. In order to examine whether the tissue is benign or malignant, a biopsy is usually performed. Unlike tumors elsewhere in the body, the biopsy of the brain tumor is not usually obtained before definitive brain surgery [8]. In order to obtain precise diagnostics, and to avoid surgery and subjectivity, it is important to develop an effective diagnostics tool for tumor segmentation and classification from MRI images [7].

The development of new technologies, especially artificial intelligence and machine learning, has had a significant impact on medical field, providing an important support tool for many medical branches, including imaging. Different machine-learning methods for image segmentation and classification are applied in MRI image processing to provide radiologists with a second opinion.

Since 2012, the Perelman School of Medicine at the University of Pennsylvania, Center for Biomedical Image Computing & Analytics (CBICA) has been running an online competition, the Multimodal Brain Tumor Segmentation Challenge (BRATS) [9]. The image databases used in BRATS are made publicly available after the competition is finished. Different classification algorithms designed using these image databases can be found in many papers [10–14]. However, the databases are usually small, on average about 285 images, and they often contain images showing two tumor levels, low and high level of glioma tumor, acquired in the axial plane [10].

In addition, classification has been carried out on other image databases, which are also quite small [15–18]. Mohsen et al. used 66 images to classify four types of images showing brain tumors: tumor-free, glioblastoma, sarcoma, and metastasis. By using a deep neural network (DNN), they obtained an accuracy of 96.97% [17].

In the literature, there are other algorithms and different modifications of the pre-trained networks that are used for image analysis, classification, and segmentation. Different approaches have been tested on other medical databases, both on MRI images of brain tumors and on tumors from different parts of the human body [19,20]. These papers were not considered further, as the focus was on the papers using the same MRI image database that we used.

Cheng et al., who were the first to present the image database used in this paper, classified the tumor types using augmented tumor region of interest, image dilatation, and ring-form partition. They extracted features using intensity histogram, gray level co-occurrence matrix, and bag-of-words models, and achieved an accuracy of 91.28% [21]. More papers that used the same database are discussed in Section 3. There, we discuss different types of networks, pre-trained ones, capsule net networks, other architectures of convolutional networks, and combinations with neural networks for feature extraction and classifiers for the output result. The discussion also concerns approaches using different modifications of the database, as well as the original one. The papers that used the original or augmented database are listed in the tables for better comparison.

The biggest problem with classifying and segmenting the MRI images with some neural networks lies in the number of images in the database. In addition, MRI images are acquired in different planes, so the option of using all the available planes could enlarge the database. As this could generally affect the classification output by overfitting, pre-processing is required before feeding the images into the neural network. However, one of the known advantages of convolutional neural networks (CNN) is that the pre-processing and the feature engineering do not have to be performed.

The aim of this research is firstly to examine the classification of three tumor types from an imbalanced database with a CNN. Although considered large compared to other available MRI image databases, this database is still far smaller than databases generally used in the field of artificial intelligence. We wanted to show that the performance of the small architecture could compare favorably with the performance of the more complex ones. Using a simpler network requires fewer resources for training and implementation. This is a crucial problem to address because limited available resources make it difficult to use the system in clinical diagnostics and on mobile platforms. If the system is needed to be used in everyday clinical diagnostics, it should be generally applicable.

We wanted to examine the network's generalization capability for clinical studies and to show how the subject-wise cross-validation approach gives more realistic results for further implementation.

In this paper, we present a new CNN architecture for brain tumor classification of three tumor types: meningioma, glioma, and pituitary tumor from T1-weighted contrast-enhanced magnetic resonance images. The network performance was tested using four approaches: combinations of two *10*-fold cross-validation methods (record-wise and subject-wise) and two databases (original and augmented). The results are presented using the confusion matrices and accuracy metric. A comparison with the comparable state-of-the-art methods is also presented.

## 2. Methodology

### 2.1. Image Database

The image database, provided as a set of slices, used in this paper contains 3064 T1-weighted contrast-enhanced MRI images acquired from Nanfang Hospital and General Hospital, Tianjin Medical University, China from 2005 to 2010. It was first published online in 2015, and the last modified version was realized in 2017 [22]. There are three types of tumors: meningioma (708 images), glioma (1426 images), and pituitary tumor (930 images). All images were acquired from 233 patients in three planes: sagittal (1025 images), axial (994 images), and coronal (1045 images) plane. The examples of different types of tumors, as well as different planes, are shown in Figure 1. The tumors are marked with a red outline. The number of images is different for each patient.

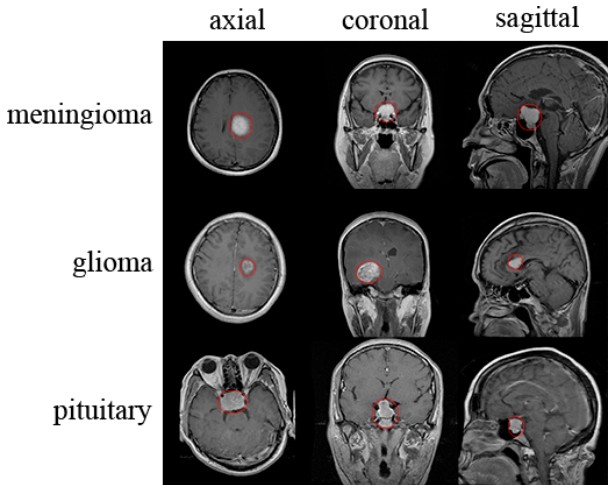

**Figure 1.** Representation of normalized magnetic resonance imaging (MRI) images showing different types of tumors in different planes. In the images, the tumor is marked with a red outline. The example is given for each tumor type in each of the planes.

### 2.2. Image Pre-Processing and Data Augmentation

Magnetic resonance images from the database were of different sizes and were provided in int16 format. These images represent the input layer of the network, so they were normalized and resized to 256 × 256 pixels.

In order to augment the dataset, we transformed each image in two ways. The first transformation was image rotation by 90 degrees. The second transformation was flipping images vertically [23]. In this way, we augmented our dataset three times, resulting in 9192 images.

### 2.3. Network Architecture

Tumor classification was performed using a CNN developed in Matlab R2018a (The MathWorks, Natick, MA, USA). The network architecture consists of input, two main blocks, classification block,

and output, as shown in Figure 2. The first main block, Block A, consists of a convolutional layer which as an output gives an image two times smaller than the provided input. The convolutional layer is followed by the rectified linear unit (ReLU) activation layer and the dropout layer. In this block, there is also the max pooling layer which gives an output two times smaller than the input. The second block, Block B, is different from the first only in the convolution layer, which retains the same output size as the input size of that layer. The classification block consists of two fully connected (FC) layers, of which the first one represents the flattened output of the last max pooling layer, whereas, in the second FC layer, the number of hidden units is equal to the number of the classes of tumor. The whole network architecture consists of the input layer, two Blocks A, two Blocks B, classification block, and output layer; altogether, there are 22 layers, as shown in Table 1.

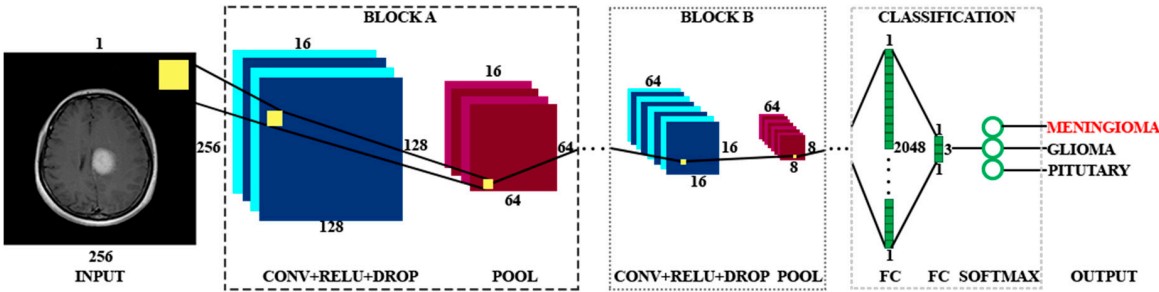

**Figure 2.** Schematic representation of convolutional neural network (CNN) architecture containing the input layer, two Blocks A, two Blocks B, classification block and output. Block A and Block B differ only in the convolution layer. Convolution layer in Block A gives an output two times smaller than the input, whereas the convolutional layer in Block B gives the same size output as input.

**Table 1.** New CNN architecture. All network layers are listed with their properties.

| Layer No. | Layer Name | Layer Properties |
|---|---|---|
| 1 | Image Input | $256 \times 256 \times 1$ images |
| 2 | Convolutional | $16\ 5 \times 5 \times 1$ convolutions with stride [2 2] and padding 'same' |
| 3 | Rectified Linear Unit | Rectified Linear Unit |
| 4 | Dropout | 50% dropout |
| 5 | Max Pooling | $2 \times 2$ max pooling with stride [2 2] and padding [0 0 0 0] |
| 6 | Convolutional | $32\ 3 \times 3 \times 16$ convolutions with stride [2 2] and padding 'same' |
| 7 | Rectified Linear Unit | Rectified Linear Unit |
| 8 | Dropout | 50% dropout |
| 9 | Max Pooling | $2 \times 2$ max pooling with stride [2 2] and padding [0 0 0 0] |
| 10 | Convolutional | $64\ 3 \times 3 \times 32$ convolutions with stride [1 1] and padding 'same' |
| 11 | Rectified Linear Unit | Rectified Linear Unit |
| 12 | Dropout | 50% dropout |
| 13 | Max Pooling | $2 \times 2$ max pooling with stride [2 2] and padding [0 0 0 0] |
| 14 | Convolutional | $128\ 3 \times 3 \times 64$ convolutions with stride [1 1] and padding 'same' |
| 15 | Rectified Linear Unit | Rectified Linear Unit |
| 16 | Dropout | 50% dropout |
| 17 | Max Pooling | $2 \times 2$ max pooling with stride [2 2] and padding [0 0 0 0] |
| 18 | Fully Connected | 1024 hidden neurons in fully connected (FC) layer |
| 19 | Rectified Linear Unit | Rectified Linear Unit |
| 20 | Fully Connected | 3 hidden neurons in fully connected layer |
| 21 | Softmax | softmax |
| 22 | Classification Output | 3 output classes, "1" for meningioma, "2" for glioma, and "3" for a pituitary tumor |

## 2.4. Training Network

We used a *k*-fold cross-validation method to test the network performance [24]. Two different approaches were implemented, and both consisted of *10*-fold cross-validation. The first approach was

to randomly divide the data into 10 approximately equal portions so that each tumor category was equally present in each portion, referred to as record-wise cross-validation. The second approach was to randomly divide the data into 10 approximately equal portions where the data from a single subject could only be found in one of the sets. Each set, therefore, contained data from a couple of subjects regardless of the tumor class, referred to as subject-wise cross-validation. The second approach was implemented to test the generalization capability of the network in medical diagnostics [25]. The generalization capability in clinical practice represents the ability to predict the diagnosis based on the data obtained from subjects from which there are no observations in the training process. Therefore, observations from individuals in the training set must not appear in the test set. If this is not the case, complex predictors can pick up a confounding relationship between identity and diagnostic status and so produce unrealistically high prediction accuracy [26]. In order to compare the performance of our network with other state-of-the-art methods, we also tested our network without *k*-fold cross-validation (one test). In all the above-mentioned methods, two data portions were used for the test, two for validation, and six for training. Both datasets, normal and augmented, were tested using all the methods.

The network was trained using an Adam optimizer, with a mini-batch size equal to 16 and data shuffling in every iteration. The early-stop condition that affects when the process of network training will stop corresponds to one epoch. More specifically, it was tuned to finish the training process after the one epoch, when the loss starts to increase. The regularization factor was set to 0.004, and the initial learning rate to 0.0004. The weights of the convolutional layers were initialized using a Glorot initializer, also known as Xavier initializer [27].

The training process was stopped when the loss on the validation set got larger than or was equal to the previous lowest loss for 11 times. The network was trained and tested on a single graphical processing unit (GPU), CUDA device, GeForce GTX 1050 Ti.

## 3. Results and Discussion

Results of the developed CNN are shown in Table 2 and visualized using the confusion matrices, as shown in Figures 3 and 5–7. In confusion matrices, non-white rows represent network output classes, and non-white columns correspond to real classes in Figures 3 and 5–7. The numbers/percentages of correctly classified images are shown on the diagonal. The last row represents the sensitivity, whereas the last column corresponds to the specificity. Overall accuracy is shown in the bottom-right field. The upper number in the non-white boxes corresponds to the number of images, and the lower number represents the percentage of the whole class database in the training or test set. In order to neglect the imbalance of classes of tumors in the database, we have also shown mean average precision, recall, and F1-score in Table 2.

**Table 2.** Results from testing the network with *10*-fold cross-validation and one test, with two different cross-validation methods and different datasets.

| Division/Dataset | Testing Approach | Test Accuracy [%] | Average Precision [%] | Average Recall [%] | Average F1-Score [%] |
|---|---|---|---|---|---|
| record-wise/the | *10*-fold | 95.40 | 94.81 | 95.07 | 94.93 |
| original dataset | One test | 97.39 | 95.44 | 96.94 | 96.11 |
| record-wise/the | *10*-fold | 96.56 | 95.79 | 96.51 | 96.11 |
| augmented dataset | One test | 97.28 | 97.15 | 97.82 | 97.47 |
| subject-wise/the | *10*-fold | 84.45 | 81.40 | 82.72 | 81.86 |
| original dataset | One test | 90.39 | 85.99 | 85.84 | 85.91 |
| subject-wise/the | *10*-fold | 88.48 | 86.48 | 87.82 | 86.97 |
| augmented dataset | One test | 91.84 | 83.94 | 82.18 | 81.78 |

Figure 3 shows confusion matrices for the record-wise *10*-fold cross-validation approach for testing data obtained from the original dataset. The classification error for the testing set after cross-validation is equal to 4.6%.

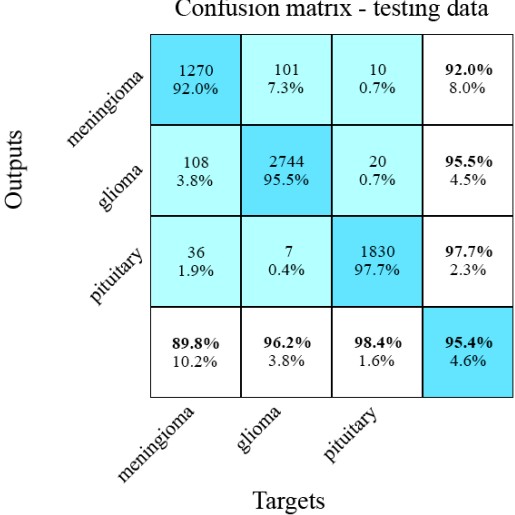

**Figure 3.** Confusion matrices for the original dataset with record-wise *10*-fold cross-validation for testing data.

Examples of classified images from the original dataset with record-wise *10*-fold cross-validation are shown in Figure 4, with the tumors outlined in red. Figure 4 comprises confusion matrices, where the rows show examples of images for the outputs, and columns emphasize what the intended target was, illustrating correctly and wrongly classified images.

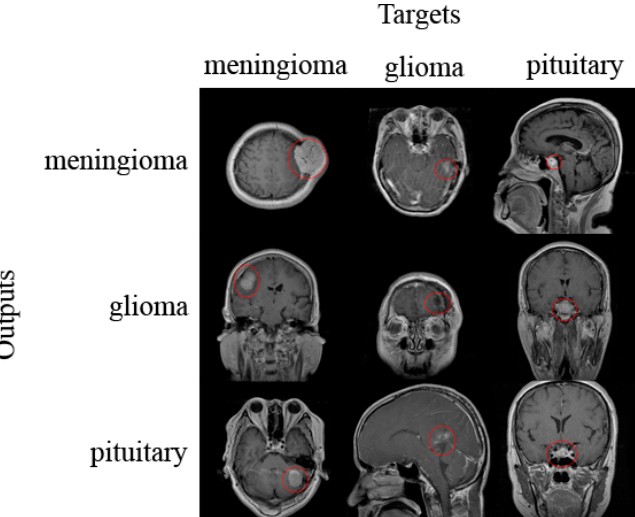

**Figure 4.** Example of classified images from the original dataset with record-wise *10*-fold cross-validation. The tumors are marked with a red outline.

Confusion matrices for the record-wise *10*-fold cross-validation method for testing data from the augmented dataset are shown in Figure 5. The classification error for the testing data is 3.4%.

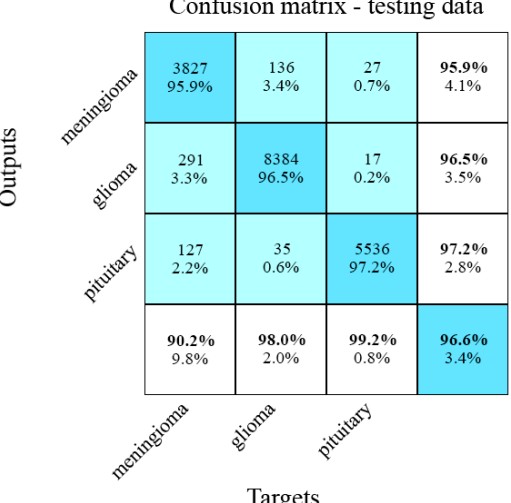

**Figure 5.** Confusion matrices for the augmented dataset with record-wise *10*-fold cross-validation for testing data.

Figure 6 presents confusion matrices for the subject-wise *10*-fold cross-validation testing method for testing data obtained from the original dataset. The classification error for the testing set is 15.7%. This result shows that the network has a smaller accuracy than for record-wise cross-validation, which was expected because the predictions were made based on previously unseen data.

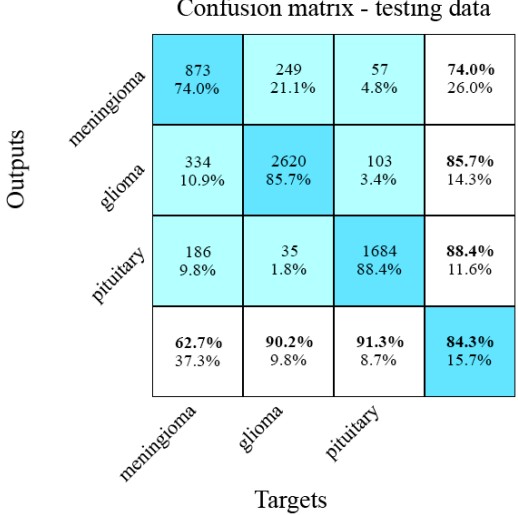

**Figure 6.** Confusion matrices for the original dataset with subject-wise *10*-fold cross-validation for testing data.

Confusion matrices for the subject-wise *10*-fold cross-validation approach for testing data from the augmented dataset are shown in Figure 7. The classification error for the testing data is 11.5%.

The proposed architecture of the CNN had only 4.3 million weights, and it obtained better results with augmented data, which was expected because the data set is not especially extensive. Even with the augmented data set, the subject-wise accuracy is lower than the accuracy obtained with the record-wise cross-validation because, with the augmentation, we only increased the number of images for individual patients, not the number of patients. As a consequence of splitting the data with the subject-wise method, increasing the number of patients was more important. The first class of tumors, meningioma, had the lowest sensitivity and specificity for all four testing methods. This is easily explained given that meningioma is the hardest to discern from the other two types of tumors based

on the place of origin and overall features. The execution speed was quite good with an average of less than 15 ms per image.

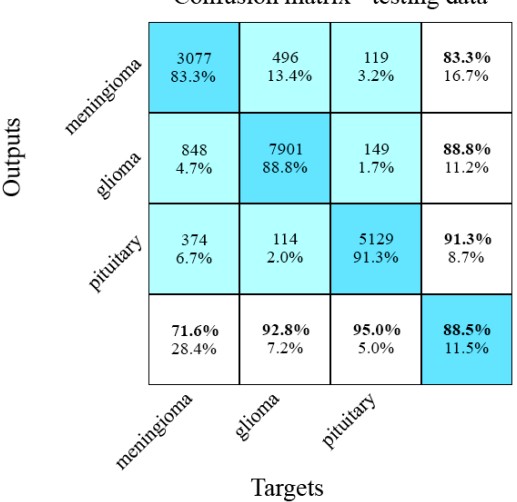

**Figure 7.** Confusion matrices for the augmented dataset with subject-wise *10*-fold cross-validation during testing.

*Comparison with State-of-the-Art-Methods*

There are several papers that use the same database for brain tumor classification. In order to compare our results with those of previous studies, we selected only those papers which had designed neural networks, used whole images as input for classification, and tested their networks with a *k*-fold cross-validation method, as shown in Table 3. We also compared our results with those of researchers who had not tested the network with *k*-fold cross-validation, as shown in Table 4. A comparison with the studies that used designed neural networks and an augmented dataset, but did not test it by *k*-fold cross-validation, is presented in Table 5.

**Table 3.** Comparison of results of different network architectures, trained and tested on the original dataset, which use whole images as input and are tested using the *k*-fold cross-validation method.

| Reference | *k*-Fold Cross-Validation Method/Data Division | Accuracy [%] | Average Precision [%] | Average Recall [%] | Average F1-Score [%] |
|---|---|---|---|---|---|
| Phaye et al. [28] | *8*-fold; data division not stated | 95.03 | X | X | X |
| Pashaei et al. [29] | *5*-fold; 80% data in training set, 20% in test | 93.68 | X | X | X |
| Gumaei et al. [30] | *5*-fold; 80% data in training set, 20% in test | 92.61 | X | X | X |
| Pashaei et al. [31] | *10*-fold; 70% data in training set, 30% in test. | 93.68 | 94.60 | 91.43 | 93.00 |
| Proposed | *10*-fold; 60% data in training set, 20% in validation, 20% in test. | 95.40 [1] | 94.81 [1] | 95.07 [1] | 94.94 [1] |

[1] The best obtained result.

**Table 4.** Comparison of results of different network architectures, trained, and tested on the original dataset, which use whole images as input and are not tested using the *k*-fold cross-validation method.

| Reference | Data Division | Accuracy [%] | Average Precision [%] | Average Recall [%] | Average F1-Score [%] |
|---|---|---|---|---|---|
| Afshar et al. [32] | data division not stated | 86.56 | X | X | X |
| Vimal Kurup et al. [33] | 80% data in training set, 20% in test | 92.60 | 92.67 | 94.67 | 93.33 |
| Srinivasan et al. [34] | 75% data in training set, 25% in test | 93.30 | X | 91.00 | 72.00 |
| Proposed | 60% data in training set, 20% in validation, 20% in test | 97.39 [1] | 95.44 [1] | 96.94 [1] | 96.11 |

[1] The best obtained result.

**Table 5.** Comparison of results of different network architectures, trained and tested on the augmented dataset, which use whole images as input and are not tested using the *k*-fold cross-validation method.

| Reference | Data Division | Accuracy [%] | Average Precision [%] | Average Recall [%] | Average F1-Score [%] |
|---|---|---|---|---|---|
| Sultan et al. [35] | 68% data in training set, 32% in validation and test | 96.13 | 96.06 | 94.43 | X |
| Proposed | 60% data in training set, 20% in validation, 20% in test | 97.28 [1] | 97.15 [1] | 97.82 [1] | 97.47 [1] |

[1] The best obtained result.

In the literature, there are also studies that used the same database for classification with pre-trained networks [23,35–40] or, as input, they use only tumor region or some features that are extracted from the tumor region [7,21,23,41,42]. Similarly, in several papers, researchers have modified this database prior to classification [36,43–47]. The designed networks are usually simpler than already-existing pre-trained networks and have faster execution speed. To our knowledge, the best results using the pre-trained network are 98.69% [40] and 98.66% accuracy [36]. Rehman et al. [40] preprocessed images with contrast enhancement and augmented the dataset. The augmentation was fivefold, with rotations of 90, 180, and 270 degrees and horizontal and vertical flipping. The best result was obtained with a fine-tuned VGG16 trained using stochastic gradient descent with momentum. Although our approach has a 1.41% higher classification error, it has 4.3 million weights as opposed to the VGG16, which is a very deep network with 138 million weights. Very deep networks such as VGG16 and AlexNet require dedicated hardware for real-time performance. Kutlu and Avcı [36] also modified the database, using only those images that were taken in the axial plane, and used only 100 images of each tumor type. For feature extraction, they used the pre-trained AlexNet, and, for testing, they performed a 5-fold cross-validation method. It is unclear how the algorithm will perform on the whole dataset and what its generalization capabilities are.

Developing the network which uses only the region of the tumor or some other segmented part as input is better in terms of speed of execution, but also requires methods for segmentation or a dedicated expert who would mark those parts.

To our knowledge, the best result in the literature using the segmented image parts as inputs are presented by Tripathi and Bag [41], with 94.64% accuracy. For input to the classifiers, they use features that are extracted from the segmented brain from the image. They tested their approach using a 5-fold cross-validation method.

## 4. Conclusions

A new CNN architecture for brain tumor classification was presented in this study. The classification was performed using a T1-weighted contrast-enhanced MRI image database which contains three tumor types. As input, we used whole images, so it was not necessary to perform any preprocessing or segmentation of the tumors. Our designed neural network is simpler than

pre-trained networks, and it is possible to run it on conventional modern personal computers. This is possible because the algorithm requires many less resources for both training and implementation. The importance of developing smaller networks is also linked to the possibility of deploying the algorithm on mobile platforms, which is significant for diagnostics in developing countries [48]. In addition, the network has a very good execution speed of 15 ms per image. In order to test the network, we used record-wise and subject-wise *10*-fold cross-validation on both the original and augmented image database. In clinical diagnostics, the generalization capability implies predictions for subjects from whom we have no observations. With this in mind, the observations from individuals in the training set must not appear in the test set. If this condition is not met, complex predictors can have unrealistically high prediction accuracy due to the confounding dependency between the identity and the diagnosis of a patient [26]. In relation to that knowledge, we have committed subject-wise cross-validation.

A comparison with the comparable state-of-the-art methods shows that our network obtained better results. The best result for *10*-fold cross-validation was achieved for the record-wise method and, for the augmented dataset, and the accuracy was 96.56%. To our knowledge, in the literature, there is no paper that shows tested generalization, through subject-wise *k*-fold method, for this image database. For the subject-wise approach, we obtained an accuracy of 88.48% for the augmented dataset. The average test execution was less than 15 ms per image. These results show that our network has a good generalization capability and good execution speed, so it could be used as an effective decision-support tool for radiologists in medical diagnostics.

Regarding further work, we will consider other approaches to database augmentation (e.g., increasing number of subjects) in order to improve the generalization capability of the network. One of the main improvements will be adjusting the architecture so that it could be used during brain surgery, classifying and accurately locating the tumor [49]. Detecting the tumors in the operating room should be performed in real-time and real-world conditions; thus, in that case, the improvement would also involve adapting the network to a 3D system [50]. By keeping the network architecture simple, detection in real time could be possible. In future, we will examine the performance of our designed neural network, as well as improved ones, on other medical images.

**Author Contributions:** Conceptualization, M.M.B.; Methodology, M.M.B. and M.Č.B.; Software, M.M.B.; Validation, M.M.B. and M.Č.B.; Writing—original draft, M.M.B.; Writing—review and editing, M.Č.B. All authors have read and agreed to the published version of the manuscript.

**Funding:** This research received no external funding.

**Acknowledgments:** We thank Vladislava Bobić and Ivan Vajs, the researchers at the Innovation Center School of Electrical Engineering, University of Belgrade, and Petar Atanasijević, the teaching assistant at the School of Electrical Engineering, for their valuable comments.

**Conflicts of Interest:** The authors declare no conflict of interest.

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
