# Peer review of "Classification of Brain Tumors from MRI Images Using a Convolutional Neural Network"

_applsci, doi:10.3390/app10061999_

Round 1

Reviewer 1 Report

The paper applies CNN on the MRI images database which consists of three classes of brain tumors (meningioma, glioma, and pituitary tumor). The structure of the paper is good. However there some main concerns about this paper.

   What is the novelty of the paper except applying an available structure (CNN) on MRI images?    Methodology Section: It is mentioned that the database is augmented three times. What is the reason to augment the dataset and compare it with the original dataset? Why the augmentation degree is chosen as three?    Methodology Section: What is the effect of each plane on the classification and the network design. Please add some explanations about that and if it is needed or not to perform some preprocessing on each plane before feeding them to CNN.    The number of images in each class of the tumor is not equal, i.e. the glioma has the highest number of samples in the dataset. How do the authors elaborate on the problem of an unbalanced dataset?    Training network Subsection: It is mentioned that the results of a k-fold validated network (subject-wise and record-wise) are compared with the state-of-the-art method with One test. Please explain which previous studies (by citing the studies) are using this method?    I would recommend explaining some information about other studies (those presented in the Results section) in the Introduction and just keep those explanations that are about the comparison between results for the results section.

Minor)

Please check the English writing grammar and structure. For example, it is recommended to add as shown in Figure. x where you are referring to a figure in the text.

Reviewer 2 Report

The manuscript “Classification of brain tumors from MRI images using convolutional neural network” presents a convolutional neural network for the classification of brain MR images in glioma, meningioma and pituitary tumors. The proposed network is validated with a benchmark dataset.

The manuscript presents a relevant problem for the CAD community but presents several weaknesses. My specific comments can be found hereafter.

Introduction

- Sentences should be supported by relevant clinical literature. I suggest asking a surgeon/clinician to check the first 2-3 paragraphs.

- A more in depth survey of the state of the art should be presented: pros and cons of each methods should be described. Otherwise, it is difficult to understand where the proposed method lies in the literature. There is already a large literature on MR image analysis with deep learning. Examples include [Litjens, Geert, et al. "A survey on deep learning in medical image analysis." Medical image analysis 42 (2017): 60-88.] and [Akkus, Zeynettin, et al. "Deep learning for brain MRI segmentation: state of the art and future directions." Journal of digital imaging 30.4 (2017): 449-459.]

- All challenges that have to be addressed when processing the MR dataset should be clearly highlighted.

- The innovation brought by the authors should be clearly outlined in the introduction. Which are the investigation hypothesis?

Methodology

- Did you manage somehow the class imbalance problem?

- Highlight in Fig. 1 the tumors (e.g., with arrows)

- Were the MR images provided as volume or set of slices? This should be clearly stated. 

- Line 89-92 and Fig. 2 could be removed because not informative

- Why did you perform only 90° rotation and flipping as augmentation? Considering that you only apply these two transformation, Fig. 3 is unuseful. Did you also perform augmentation on the fly during training? 

- The rationale behind the proposed architecture is unclear. How did you define the number of layers? Ablation studies are required: how does the performance of the CNN vary when adding/removing layers? 

- Line 129-129: repetition

- What is the early stopping parameter?

- An “experiments” section should be added to list the comparison methods and the performance metrics that will be used to evaluate the results.

- In Table 2, I really do not see the sense of comparing k-fold and hold-out cross validation results.

- Avoid reporting confusion matrices for training data. 

- In Table 3-4-5, how can you say that you performed the best if the testing data are different?  Considering that the testing dataset is not balanced, why did you use accuracy as performance metric? I would suggest using a more appropriate metric. 

- The discussion should be improved by interpreting the significance of your findings in light of what was already known in the literature and by explaining any new understanding that emerged from your work.

- It would be nice to see visual samples of wrongly classified images.

Conclusion

The future work paragraph is rather poor. For example, you could also cite approaches to brain-tumor surgery, which may benefit from using the proposed methodology for accurately locate, via image registration, the tumor in the operating room. (e.g., Moccia, Sara, et al. "Toward improving safety in neurosurgery with an active handheld instrument." Annals of biomedical engineering 46.10 (2018): 1450-1464.). Furthermore, you could also explore 3D CNNs, which have already shown promising results in similar fields (e.g., Zaffino P, Pernelle G, Mastmeyer A, Mehrtash A, Zhang H, Kikinis R, Kapur T, Spadea MF. Fully automatic catheter segmentation in MRI with 3D convolutional neural networks: application to MRI-guided gynecologic brachytherapy. Physics in Medicine & Biology. 2019 Aug 14;64(16):165008.)

Round 2

Reviewer 1 Report

Thank you. I don't have any further questions.

Author Response

The authors would like to thank the reviewer. 

Reviewer 2 Report

I would like to thank the authors for revising the manuscript “Classification of brain tumors from MRI images using convolutional neural network”.  

Considering the first comment, I may understand the authors’ point. However, the authors are dealing with the medical field. It would be better to shorten the clinical paragraph to avoid writing inaccurate sentences.

As for the MRI challenges, I appreciate the authors’ effort. However, the listed challenges are not specific to MRI. The authors should make an effort to identifying challenges relevant to the addressed topic.

The response to “Did you manage somehow the class imbalance problem?” actually does not answer my question. 

Instead of writing early-stop parameter, I would say early-stop condition.

It is still not clear to me why the authors performed both hold out and k-fold cross validation.
